# First stage progression in women with spontaneous onset of labor: A large population-based cohort study

Louise Lundborg[1]*, Katarina Åberg[1], Anna Sandström[1,2,3,4], Andrea Discacciati[5,6], Ellen L. Tilden[4,7], Olof Stephansson[1,2], Mia Ahlberg[1,2]

1 Clinical Epidemiology Division, Department of Medicine, Solna, Karolinska Institutet, Stockholm, Sweden, 2 Division of Obstetrics and Gynecology, Department of Women´s and Children´s Health, Karolinska University Hospital and Institutet, Stockholm, Sweden, 3 Department of Women's and Children's Health, Uppsala University, Uppsala, Sweden, 4 Department of Obstetrics and Gynecology, Oregon Health & Science University School of Medicine, Portland, Oregon, United States of America, 5 Unit of Biostatistics, Institute of Environmental Medicine, Karolinska Institutet, Stockholm, Sweden, 6 Department of Medical Epidemiology and Biostatistics, Karolinska Institutet, Stockholm, Sweden, 7 Department of Nurse-Midwifery, Oregon Health & Science University School of Nursing, Portland, Oregon, United States of America

* louise.lundborg@ki.se

**Data Availability Statement:** The Stockholm-Gotland Obstetric Cohort was used for this study. Information in the database was retrieved from the medical record system Obstetrix. The regional ethical committee at Karolinska Institutet,

## Abstract

### Objective

To describe the duration, progression and patterns of first stage of labor among Swedish women.

### Design

Population-based cohort study.

### Population

Data from Stockholm-Gotland Obstetric Cohort 2008–2014 including ¼ of all births in Sweden, the final sample involved a total of 85,408 women with term, singleton, vertex, live fetuses experiencing spontaneous labor onset and vaginal delivery with normal neonatal outcomes.

### Main outcome measures

Time to progress during first stage of labor using three approaches: 1) Traverse time in hours to progress centimeter to centimeter, 5th, 50th (and 95th percentile); 2) Dilation curves for different percentiles, and; 3) Cumulative duration for the 95th percentile by parity and dilation at admission.

### Results

Variation in both the total duration and the trajectory of cervical change over time is large. Similar to the general held view, the rate of cervical dilation accelerates at 5–6 centimeters. Among nulliparous women, the median time found in our population was faster than their

Stockholm, Sweden approved the study protocol (No 2009/275-31, 2012/365-32, 2013/792-32, 2014/177-32, 2014/962-32). The database is stored in the Unit of Clinical epidemiology at Karolinska Institutet Stockholm, Sweden. Public datasharing from this data base is not permitted. Any questions regarding public access to the data is handled by Unit of Clinical epidemiology. Department of Medicine, Karolinska Institutet, Professor sven.cnattingius@ki.se@ki.se.

**Funding:** This study was supported by grants provided by the Stockholm County Council (ALF project 2017-01000 MA) https://forskningsstod. vmi.se/Ansokan/start.asp, and Swedish Research Council for Health, Working Life and Welfare (STYA-2017/0003) (MA, LL) https://www. government.se/government-agencies/swedish-research-council-for-health-working-life–forskningsradet-for-arbetsliv-halsa-och-valfard-forte/ The funding sources had no role in study design, collection of data, preparation of the manuscript, analysis or interpretation of data, nor in decision to submit the article for publication.

**Competing interests:** Competing interests: The authors declare no competing interests.

counterparts in studies conducted on American and African cohorts. Among nulliparous and multiparous women our data suggest that the median cervical change over time is faster than 1 cm per hour during the first stage of labor. However, traverse time of cervical change at and beyond the 95th percentile is longer than 1 cm per hour.

## Conclusions

Labor progression varies widely and labors experiencing a prolonged first stage can still result in normal outcomes. The assumption of 1 cm per hour cervical dilation rate for the first stage of labor may not be universally meaningful. There are differences in progression for women during first stage of labor in different populations. For prolonged labor progression to be more clinically meaningful, the association with adverse birth outcomes needs to be further investigated in specific populations.

## Introduction

Friedman introduced the first graphic analysis of labor progress in the 1950's [1, 2]. Based on the Friedman labor curve, Philpott and colleagues [3, 4] developed guidelines to monitor labor and detect deviating labor progression. Influenced by the work of both Friedman [1, 2, 5, 6] and Philpott [3, 4] the World Health Organization (WHO) partograph was constructed [7]. The partograph's alert line represent cervical dilation of 1 cm per hour and an action line is placed after the alert line, usually after two to four hours. The assumption that normal labor progression is linear has had a huge impact on labor management globally over the last 50 years. Several actions have been made to expedite birth when labor continues longer than thresholds deemed 'labor dystocia' by this traditional partograph [8–11]. The cesarean delivery rate has increased worldwide, and one major reason for this increase is cesarean delivery due to slow labor—labor dystocia [12–16]. Sweden has a similar increase but differs with an overall lower rate, approximately 18% of deliveries are via cesarean, compared with many other high-income countries [17–20]. Over the last decade, both the strict timelines recommended by the traditional partograph and the shape of the labor curve have been challenged [21–27]. In 2010, Zhang and colleagues presented a hyperbolic labor curve with faster progression after 6 cm, suggesting that some cesarean deliveries due to labor dystocia early in labor could be prevented [26]. Further examination of this question in other populations and with large data sets is needed to advance this line of inquiry. Swedish healthcare systems routinely collect granular, population level data. This presents a unique opportunity to conduct epidemiological maternity care research.

We conducted a large (n = 175,522) population-based register cohort study, using the same selection criteria and similar statistical methods as two recent studies on contemporary populations conducted by Zhang et al. [26] (n = 62,415, American cohort) and Oladapo et al. [23] (n = 5606, African cohort). The purpose of this study was to refine this emerging contemporary understanding of "normal" labor progression through describing: 1) the duration of the first stage labor, evaluating both duration to progress from one centimeter of cervical dilation to the next as well as the total duration, and 2) trajectories of cervical dilation by various quantiles in Swedish women with term, singleton, vertex, live fetuses experiencing spontaneous labor onset and vaginal delivery with normal neonatal outcomes.

## Material and methods

Data were obtained from the population-based Stockholm-Gotland Obstetric Cohort, a data base that captures information directly from the electronic medical record system Obstetrix on about 25% of all pregnancies and deliveries in Sweden. This data base includes prospectively collected variables regarding maternal characteristics, maternal medical history, processes and care from pregnancy through birth, as well as neonatal health and care. All care in this setting is free of charge during pregnancy and childbirth, and more than 99.9% of all women give birth in a hospital. In this setting, antenatal care is offered to all women. The first visit generally takes place between 7 to 12 weeks of gestation, with approximately 10–12 visits during pregnancy depending on parity. Midwives are the primary caregivers in Sweden, and women are referred to obstetricians when needed. The Stockholm-Gotland Obstetric database further contains granular information regarding the onset of labor, labor management including detailed information from the partograph such as time points for cervical dilation, cervical exams, epidural analgesia use, and oxytocin for labor augmentation, mode of delivery, infant characteristics, and maternal and neonatal diagnoses according to the International Classification of Diseases, 10th revision (ICD-10).

Between January 2008 and October 2014, data regarding 175,522 pregnancies and births was collected. We included 85,408 women with term (37+0 until 41+6 weeks of gestation), vertex, singleton pregnancies who experienced spontaneous labor onset with a live fetus. We excluded women whose labors were induced or who delivered via cesarean section or had a previous cesarean delivery. We also excluded women with less than two cervical examinations during first stage and records in which timing of the complete dilation of cervix was missing to be able to create labor curves in a reversed approach from fully dilated cervix. As our intent was to describe labor processes among those with uncomplicated outcomes, we also excluded births when neonates had an Apgar score less than 7 at 5 minutes of age or any of the following morbidities assessed by ICD-10 diagnostic codes: hypoxic ischæmic encephalopathy (HIE) (P916A, P916B, P916C, P916X), convulsion (P909, P909A, P909B, P909C) and meconium aspiration (P240). Fig 1 describes the sample selection process for the study cohort.

With focusing on nulliparous women, the start of first stage of labor was defined by two strict criterions: C1) time characterized by well effaced cervix (complete or nearly 100% effacement), cervical dilation of ≥3 centimeter in presence of regular painful uterine contractions or alternatively C2) time characterized by well effaced cervix (complete or nearly 100% effacement), a rupture of membrane in combination with cervical dilation of ≥3 centimeter. Following this hierarchy of the two criterions, when there were two or three time points fulfilled in one criterion, the second time point was chosen as the starting point of first stage of labor. If a woman did not fulfill all the parameters of criterion one, she was evaluated according to criterion two. Women who did not fulfill criterion one or two were not included in the study cohort since the onset of the first stage of labor could not be clearly defined. These criterions are based on the Swedish standard practice to identify women who are in first stage of labor and used as a guideline at admission at all obstetric units in the cohort [28]. The termination of the first stage was defined by the time-point that the cervix was fully dilated. The estimated median duration of the first stage of labor was extrapolated from the cervical examination on admission and subsequent examinations performed during labor. Labor duration prior to hospital admission was not considered in this analysis.

To explore potential selection bias, we also conducted analyses with a cohort including women delivered via cesarean section and adverse neonatal outcomes. Our intent with this step in the analysis was to describe characteristics, first stage duration and the trajectory of cervical change among women regardless mode of delivery or adverse birth outcome. In this

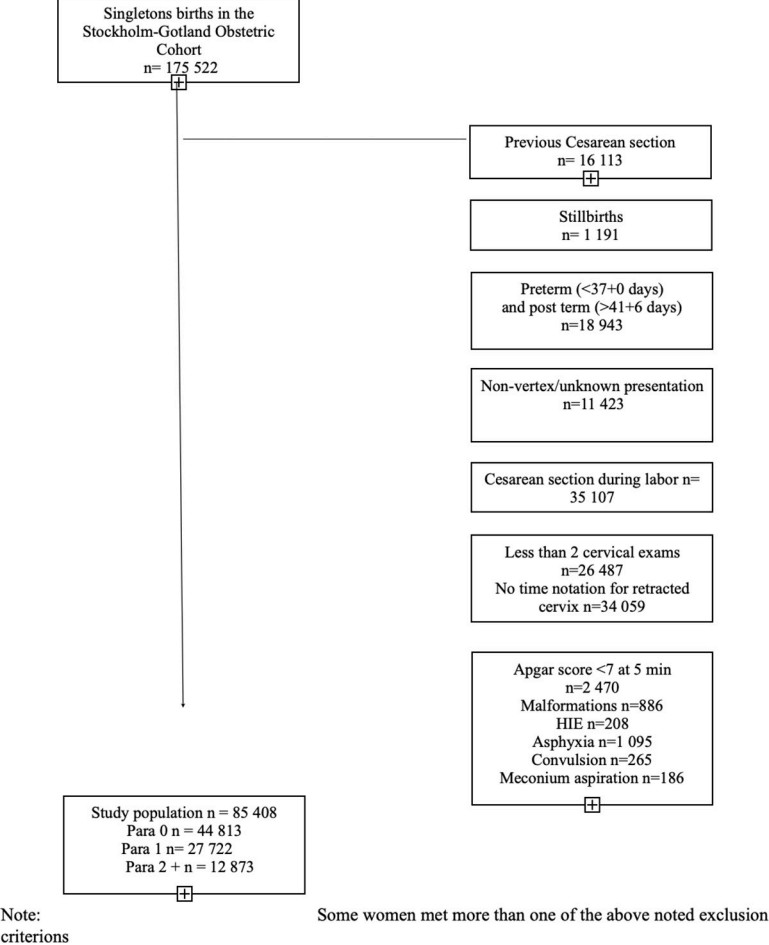

**Fig 1. Diagram of patient selection for study cohort.** Gestational age was determined using the following hierarchy: (a) embryo transfer, (b) first trimester ultrasound (c) early second trimester ultrasound offered to all women, (d) date of last menstrual period and (e) postnatal assessment. Characteristics of the study populations are reported as frequencies and percentages for categorical variables and as means and standard deviations or as medians and 10th, 90th percentiles for continuous variables, as indicated (Table 1).

target population cohort, we included 101 730 women with term (37+0 until 41+6 weeks of gestation), vertex, singleton pregnancies who experienced spontaneous labor onset with a live fetus.

The starting point of first stage was defined identical as for the above described cohort. The endpoint was either time point for fully dilated cervix or time point for cesarean delivery.

S1 Fig describes the sample selection process for the target population cohort.

## Statistical analysis

Three approaches were used to describe the duration of first stage of labor and the trajectory of cervical change: 1) Traverse time in hours to progress centimeter to centimeter, 5th, 50th (median) and 95th percentile; 2) Dilation curves for different percentiles of cervical dilation, and; 3) 95th percentile of cumulative duration of labor stratified by parity and dilation at admission.

Women were categorized into three parity groups (0,1, and 2+) to explore any differences according to parity. Labor duration and the trajectory of cervical change were characterized at

every centimeter, starting at 3-centimeter dilation, during the first stage of labor. To estimate the distributions of traverse times, which are the duration to progress from one cm of cervical dilation to the next, we used interval censored regression under the assumption that traverse times followed log-normal distributions [29].

To construct labor dilation curves, the earliest recorded time of full cervical dilation was considered as the starting point or time zero, to anchor the curves. It was performed in an analysis similar to previous studies and time were calculated backwards: the axis was reverted to a positive value after computation [23, 25, 26]. Time points of cervical dilation were expressed as hours prior to the time of full cervical dilation. For example, if a measurement was performed 45 minutes prior to full cervical dilation, then time = 0.75 hours [29]. Since we were not only interested in mean cervical dilation duration, but rather in its distribution, we used quantile regression to estimate a set of quantiles (10th, 25th, 50th, 75th, 90th, and 95th) of cervical dilation conditional on time to complete cervical dilation. Of note, quantile regression does not make any assumptions about the conditional distribution of the outcome (here, cervical dilation) given covariates (here, time to complete dilation) Furthermore, since cervical dilation is a measure bounded between 0 and 10 cm, we performed quantile regression on a logit transform of cervical dilation. This method, which hinges on the fact that quantiles are invariant to monotone transforms, such as the logit, is known as logistic quantile regression [30]. Time to complete cervical dilation, the only covariate of our models, was modelled using restricted cubic splines with three knots [31].

To evaluate any presumptive differences in the duration of and trajectory of cervical change, we also estimated cumulative labor stratified by cervical dilation at admission and starting from 3, 4, 5, or 6 cm. This was done using an analogous interval-censored regression approach to the one used for traverse time estimation (Table 3, Figs 2–4). All statistical analyses were performed using Stata 14.2 (StataCorp, College Station, TX, USA). The regional ethical committee at Karolinska Institutet, Stockholm, Sweden approved the study protocol (No 2009/275-31, 2012/365-32, 2013/792-32, 2014/177-32, 2014/962-32). Written informed consent was not required by the ethical committee. All data were anonymized prior to access. The database is stored in the Unit of Clinical epidemiology at Karolinska Institutet Stockholm, Sweden.

## Results

Table 1 presents the characteristics of women and infants by parity. The majority of women had a spontaneous vaginal birth (nulliparous = 81.4%; parity 1 = 96.9%; parity 2+ = 96.5%). Nulliparous women were younger than those with one and two-plus parity, and composed 52% of the sample. Mean body mass index (BMI) increased by one unit with each advancing level of parity (23 vs. 24 vs. 25). Oxytocin for augmentation was used in almost half of the nulliparous women (49.1%), and use decreased to 24.1% in parity 1 and slightly less, 23.7%, in parity 2+. Epidural analgesia was used in 64.5% of the nulliparous women, 32.6% of women with parity 1, and 22.9% of women with parity 2+. Mean birth weight increased by about 150 grams from first to second-born infants but only by another 26 grams from the second-born to subsequent infants. Mean gestational age was similar across parity groups.

The estimated 5th, 50th, and 95th percentiles of traverse times' distributions, i.e. duration of labor from one centimeter to the next, using 3 cm as a starting point, are reported in Table 2. Traverse time in the 95th percentile well exceeds 1 cm per hour at all stages in all parity groups. However, the 50th percentile in both nulliparous and multiparous shows faster progression than 1 cm per hour at all stages. To support clinical translation, we also generated graphic representations that depict a random sample of numerous women's labor curves, by parity and based on percentiles rather than average labor curves (Figs 2–4).

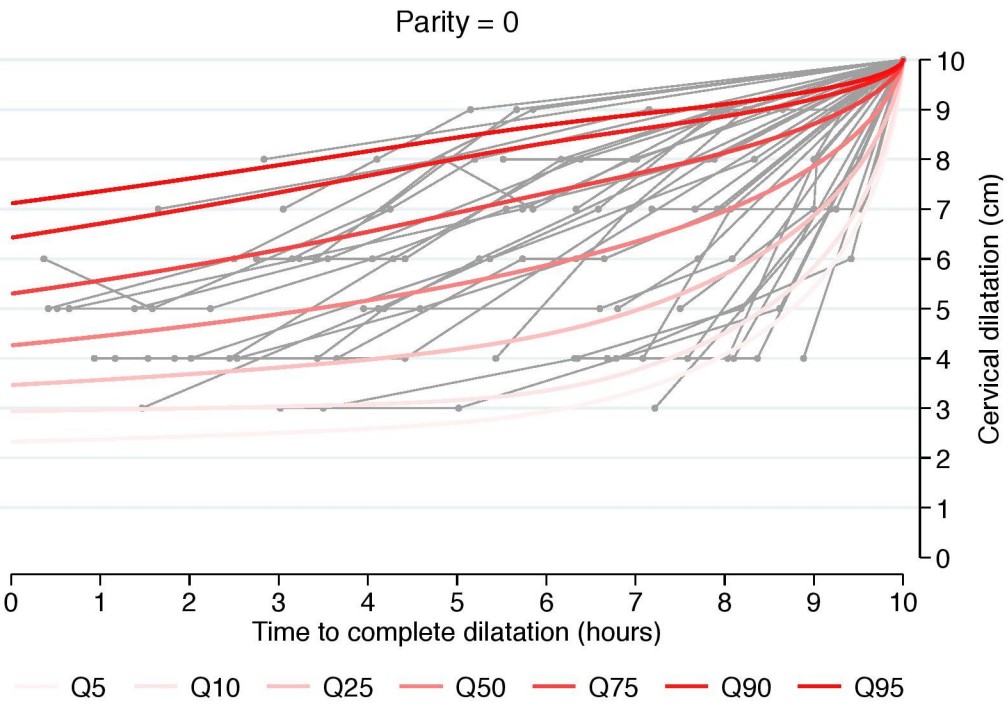

**Fig 2. Labor curve for a random sample of numerous women based on percentiles 5-95th for nulliparous.** Labor curves with a reversed approach from fully dilated cervix in singleton term pregnancies with spontaneous onset of labor and vertex presentation, vaginal delivery and healthy neonates.

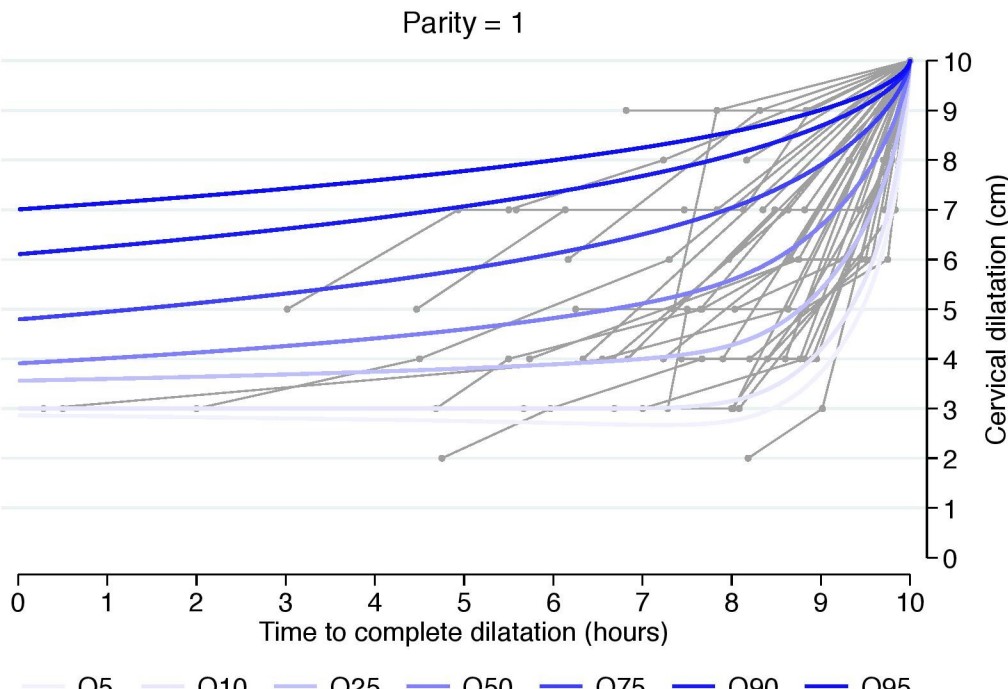

**Fig 3. Labor curve for a random sample of numerous women based on percentiles 5-95th for parity = 1.** Labor curves with a reversed approach from fully dilated cervix in singleton term pregnancies with spontaneous onset of labor and vertex presentation, vaginal delivery and healthy neonates.

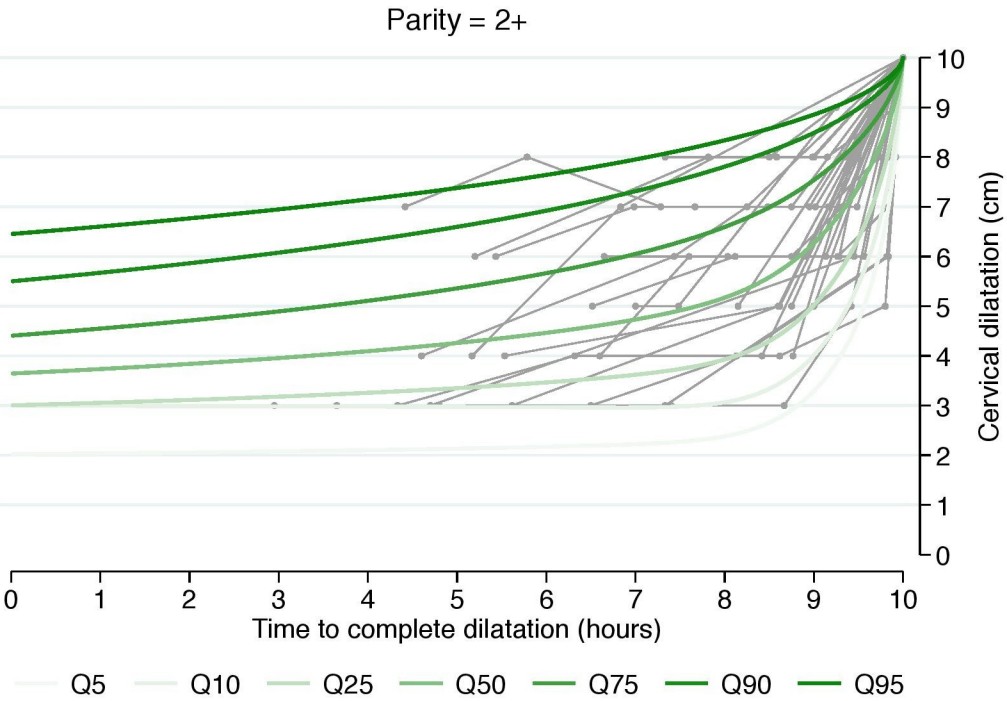

**Fig 4. Labor curve for a random sample of numerous women based on percentiles 5-95th for parity = 2+.** Labor curves with a reversed approach from fully dilated cervix in singleton term pregnancies with spontaneous onset of labor and vertex presentation, vaginal delivery and healthy neonates.

Table 3 describes the cumulative duration of labor of women in the 5th, 50th, and 95th percentiles. Results show that the estimated cumulative duration of time includes wide variation, consistent with the data previously displayed in Table 2. While those whose first stage of labor progression at the 50th percentile showed cervical dilation trajectories of 1 cm per hour in all parity groups, those with first stage of labor progression at the 95th percentile who were admitted at 3 cm showed a substantially longer first stage, lasting for >16 hours among nulliparous women and >11 hours among multiparous women. For women admitted at 4 cm, first stage duration among nulliparous women could last >14 hours and among multiparous women first stage duration was slightly less than 10 hours.

Figs 5–7 illustrate the 95th percentiles of cumulative labor duration, plotted as staircase lines based on the cervical dilation observed at admission, and centimeter by centimeter until full cervical dilation. The staircase cumulative duration is equal to the 95th percentiles in Table 3.

S1 Table presents the characteristics of women and infants by parity in the target population cohort. The majority of women regardless of parity had a spontaneous vaginal birth. Among nulliparous 70.9% had a spontaneous vaginal birth and 11,4% (6 451 women) had a cesarean delivery. For parity 1 94,5% had a spontaneous vaginal birth and 2.1% (604 women) had a cesarean delivery. Posterior fetal position was 6.1% for nulliparous, 4,2% for parity 1 and 3,8% for parity 2+. The estimated 5th, 50th, and 95th percentiles of traverse times' distributions, i.e. duration of labor from one centimeter to the next are reported in S2 Table. Traverse time in the 95th percentile well exceeds 1 cm per hour at all stages in all parity groups and the estimated duration exceeds the corresponding distributions in the study cohort. S3 Table describes the cumulative duration of labor of women in the 5th, 50th, and 95th percentiles. The estimated cumulative duration of time includes a wide variation, consistent with the previously described data from the study cohort. In the target cohort duration of first stage among

**Table 1. Baseline characteristics of the study population by parity.**

| Characteristics | Parity = 0 n = 44 813 | % | N | Parity = 1 n = 27 722 | % | N | Parity = 2+ n = 12 873 | % | N |
|---|---|---|---|---|---|---|---|---|---|
| **Age** years, mean (SD) | 29.2 (4.96) | | 44 783 | 31.6 (4.65) | | 27 707 | 34.1 (4.63) | | 12 872 |
| **Maternal height cm** mean (SD) | 166.6 (6.43) | | 44 358 | 166.4 (6.48) | | 27 480 | 165.5 (6.48) | | 12 754 |
| **BMI.** kg/m$^2$ at first visit antenatal clinic | 23 (3.94) | | 43 132 | 24 (4.35) | | 26 666 | 25 (4.68) | | 12 424 |
| **Family situation** | | | | | | | | | |
| Single | | 2.3 | 1 044 | | 1.2 | 323 | | 2.4 | 305 |
| Co-habitant | | 90.3 | 40 454 | | 95.2 | 26 403 | | 92.5 | 11 902 |
| Missing | | 7.4 | 3 315 | | 3.6 | 994 | | 5.2 | 666 |
| **Amniotic membranes status at admisson** | | | | | | | | | |
| Intact | | 65.8 | 29 494 | | 75.6 | 20 946 | | 76.4 | 9 840 |
| Ruptured | | 32.2 | 14 443 | | 22.6 | 6 259 | | 21.7 | 2 798 |
| Missing | | 2.0 | 876 | | 1.9 | 517 | | 1.8 | 235 |
| **Cx* dilation at admission** median 10$^{th}$,90$^{th}$) | 5 (3,8) | | 44 813 | 5 (3,8) | | 27 722 | 5 (3,8) | | 12 873 |
| **Cx* exams during labor** median (10$^{th}$,90$^{th}$) | 4 (2,7) | | 44 813 | 3 (2,7) | | 27 722 | 3 (2,5) | | 12 873 |
| **Oxytocin** (%) | | | | | | | | | |
| No | | 50.9 | 22 830 | | 78.6 | 21 779 | | 76.3 | 9 817 |
| Yes | | 49.1 | 21 983 | | 21.4 | 5 943 | | 23.7 | 3 056 |
| **Epidural** (%) | | | | | | | | | |
| No | | 35.5 | 15 896 | | 67.4 | 18 688 | | 77.1 | 9 924 |
| Yes | | 64.5 | 28 917 | | 32.6 | 9 034 | | 22.9 | 2 949 |
| **Mode of birth** (%) | | | | | | | | | |
| Non-instrumental vaginal birth | | 81.4 | 36 465 | | 96.9 | 26 865 | | 96.5 | 12 427 |
| Instrumental delivery | | 18.6 | 8 348 | | 3.1 | 857 | | 3.5 | 446 |
| **Gestational length at birth** mean (SD) | 40.1 (1.09) | | 44 813 | 40.0 (1.07) | | 27 722 | 39.9 (1.12) | | 12 873 |
| **Birth weight in grams** mean (SD) | 3 474 (437.8) | | 44 788 | 3 640 (460.37) | | 27 709 | 3 666 (483.13) | | 12 868 |
| **Head circumference in cm** mean (SD) | 34 (1.73) | | 44 648 | 35 (1.74) | | 27 656 | 35 (1.68) | | 12 835 |
| **Fetal position** (%) | | | | | | | | | |
| Occiput anterior | | 96.8 | 43 390 | | 96.3 | 27 704 | | 96.5 | 12 427 |
| Occiput posterior | | 3.2 | 1 423 | | 3.7 | 1 018 | | 3.5 | 446 |

nulliparous are longer than in the study cohort. S2–S4 Figs illustrate the 95$^{th}$ percentiles of cumulative labor duration for the target population cohort and plotted as staircase lines based on the cervical dilation observed at admission, and centimeter by centimeter until full cervical

**Table 2. Duration of labor in hours from one-centimeter dilation to the next, by parity.**

| Cervical dilation cm | Cervical dilation cm | Parity 0 Duration in hours (min-max) | n | Parity 1 Duration in hours (min-max) | n | Parity 2+ Duration in hours (min-max) | n |
|---|---|---|---|---|---|---|---|
| 3 | 4 | 0.83 (0.14–4.97) | 8 876 | 0.47 (0.05–4.65) | 3 566 | 0.52 (0.05–5.05) | 2 152 |
| 4 | 5 | 0.89 (0.15–5.26) | 17 425 | 0.38 (0.04–3.82) | 7 550 | 0.43 (0.04–4.62) | 4 113 |
| 5 | 6 | 0.68 (0.10–4.55) | 19 017 | 0.23 (0.02–2.90) | 8 962 | 0.19 (0.01–3.23) | 4 454 |
| 6 | 7 | 0.48 (0.06–3.82) | 17 969 | 0.12 (0.01–2.25) | 8 887 | 0.10 (0.01–2.37) | 4 135 |
| 7 | 8 | 0.33 (0.03–3.62) | 16 599 | 0.08 (0.00–1.94) | 8 470 | 0.05 (0.00–1.85) | 3 717 |
| 8 | 9 | 0.24 (0.02–3.00) | 16 326 | 0.04 (0.00–1.58) | 8 233 | 0.02 (0.00–1.27) | 3 500 |
| 9 | 10 | 0.18 (0.01–2.66) | 14 611 | 0.02 (0.00–1.18) | 5 990 | 0.01 (0.00–1.03) | 2 357 |

Data reported as median hours (5$^{th}$, 95$^{th}$ percentiles)

**Table 3. Cumulative duration of labor in hours in para 0, 1, and 2+ based on the cervical dilation at admission.**

| From cervical dilation | To cervical dilation | Parity 0 | | 1 | | 2+ | |
|---|---|---|---|---|---|---|---|
| | | Duration in hours | N | Duration in hours | N | Duration in hours | N |
| 3 | 4 | 0.81 (0.14, 4.88) | 7 694 | 0.47 (0.05, 4.50) | 3 220 | 0.53 (0.06, 4.70) | 1 893 |
| 3 | 5 | 2.66 (0.78, 8.80) | 7 661 | 1.66 (0.35, 7.90) | 3 217 | 1.80 (0.39, 8.46) | 1 895 |
| 3 | 6 | 3.89 (1.38, 10.98) | 7 692 | 2.48 (0.65, 9.55) | 3 222 | 2.56 (0.69, 9.80) | 1 900 |
| 3 | 7 | 4.66 (1.72, 12.62) | 7 689 | 2.88 (0.80, 10.35) | 3 225 | 2.95 (0.81. 10.68) | 1 903 |
| 3 | 8 | 5.27 (1.98, 14.04) | 7 678 | 3.15 (0.91, 10.94) | 3 223 | 3.15 (0.88, 11.23) | 1 906 |
| 3 | 9 | 5.78 (2.20, 15.13) | 7 701 | 3.35 (0.99, 11.38) | 3 225 | 3.31 (0.95, 11.51) | 1 906 |
| 3 | 10 | 6.21 (2.38, 16.23) | 7 717 | 3.47 (1.03,11.70) | 3 228 | 3.39 (0.97, 11.77) | 1 906 |
| 4 | 5 | 0.89 (0.16, 5.16) | 11 442 | 0.38 (0.04, 3.82) | 5 711 | 0.44 (0.04, 4.56) | 2 890 |
| 4 | 6 | 2.29 (0.63, 8.32) | 11 442 | 1.18 (0.22,6.37) | 5 710 | 1.23 (0.21, 6.95) | 2 891 |
| 4 | 7 | 3.27 (1.05, 10.18) | 11 452 | 1.77 (0.41, 7.70) | 5 715 | 1.78 (0.40,8.03) | 2 897 |
| 4 | 8 | 3.94 (1.34, 11.65) | 11 473 | 2.09 (0.51, 8.56) | 5 710 | 2.08 (0.50,8.60) | 2 900 |
| 4 | 9 | 4.52 (1.58, 12.94) | 11 488 | 2.33 (0.59,9.24) | 5 714 | 2.27 (0.57, 9.02) | 2 904 |
| 4 | 10 | 4.99 (1.76, 14.11) | 11 520 | 2.48 (0.63,9.69) | 5 720 | 2.38 (0.60,9.44) | 2 905 |
| 5 | 6 | 0.66 (0.09, 4.61) | 8 150 | 0.21 (0.02, 2.81) | 5 725 | 0.21 (0.01, 3.31) | 2 363 |
| 5 | 7 | 1.66 (0.38, 7.35) | 8 149 | 0.69 (0.09, 5.00) | 5 215 | 0.67 (0.08, 5.67) | 2 366 |
| 5 | 8 | 2.51 (0.67, 9.35) | 8 161 | 1.13 (0.20, 6.93) | 5 220 | 1.10 (0.18, 6.92) | 2 367 |
| 5 | 9 | 3.15 (0.90, 10.92) | 8 163 | 1.49 (0.29, 7.13) | 5 216 | 1.38 (0.25, 7.62) | 2 368 |
| 5 | 10 | 3.65 (1.08, 12.33) | 8 201 | 1.59 (0.33, 7.66) | 5 224 | 1.50 (0.28, 8.14) | 2 374 |
| 6 | 7 | 0.49 (0.05, 3.81) | 5 783 | 0.11 (0.01,2.20) | 5 232 | 0.10 (0.04,2.22) | 1 864 |
| 6 | 8 | 1.22 (0.22, 6.77) | 5 779 | 0.46 (0.05,4.29) | 4 449 | 0.40 (0.04, 4.20) | 1 861 |
| 6 | 9 | 2.09 (0.49, 8.93) | 5 800 | 0.84 (0.13, 5.65) | 4 457 | 0.70 (0.09,5.28) | 1 867 |
| 6 | 10 | 2.67 (0.67, 10.70) | 5 843 | 1.05 (0.17, 6.36) | 4 466 | 0.85 (0.12, 5.90) | 1 870 |

dilation. Including women with cesarean delivery in the target population cohort did not lead to any major change in pattern or trajectories for parity 1 and parity 2+ women.

## Discussion

### Swedish cohort vs Friedman and contemporary cohorts

There is wide variation of first stage labor progression in this low-risk population, and more rapid labor progress should not be expected until at least 5–6 centimeter of cervical dilation regardless of parity. These findings correspond to previous studies by Zhang [26] and Oladapo [23], displayed in Table 4.

Interestingly, in the Swedish cohort, the median progression from one centimeter to the next was more rapid than one hour from 3 cm dilation and throughout first stage. These results differ from the results of both Zhang [26] and Oladapo [23]and is even shorter than the 1 cm per hour average cervical dilation described in the landmark publications of Friedman and others [1, 2, 5]. Results also suggest that women with the longest labors can progress slowly during the entire first stage. For example, it might take as long as 3 hours to progress from 8 to 9 cm dilation for a woman giving birth to her first child. Those whose labor durations reached or exceeded the 95[th] percentile had substantially longer labor duration after 6 cm compared to Zhang's findings [26] but our results were similar to Oladapo's findings [23]. Findings of this analysis support previous authors' conclusions that an "average" dilation median rate for women during the first stage of labor does not exist [22, 23, 32, 33].

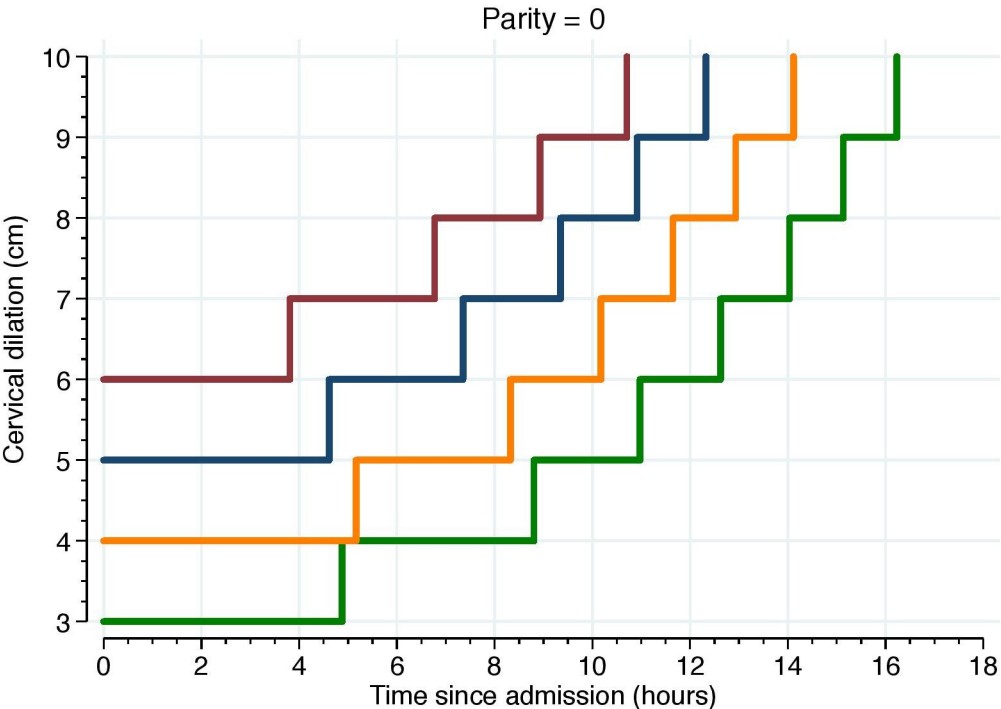

**Fig 5. The 95th percentiles of cumulative duration among nulliparous.** Illustrate the 95th percentiles of cumulative labor duration in parity = 0 and plotted as staircase lines based on the cervical dilation observed at admission, and centimeter by centimeter until full cervical dilation. The staircase cumulative duration is equal to the 95th percentiles in Table 3.

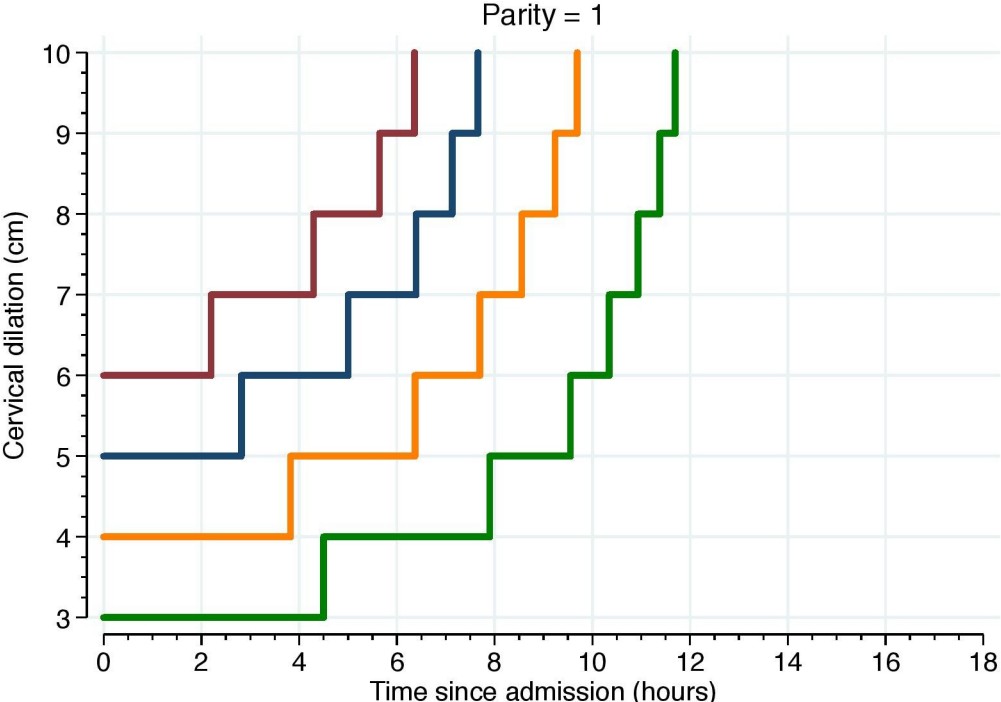

**Fig 6. The 95th percentiles of cumulative duration among parity = 1.** Illustrate the 95th percentiles of cumulative labor duration in parity = 1 and plotted as staircase lines based on the cervical dilation observed at admission, and centimeter by centimeter until full cervical dilation. The staircase cumulative duration is equal to the 95th percentiles in Table 3.

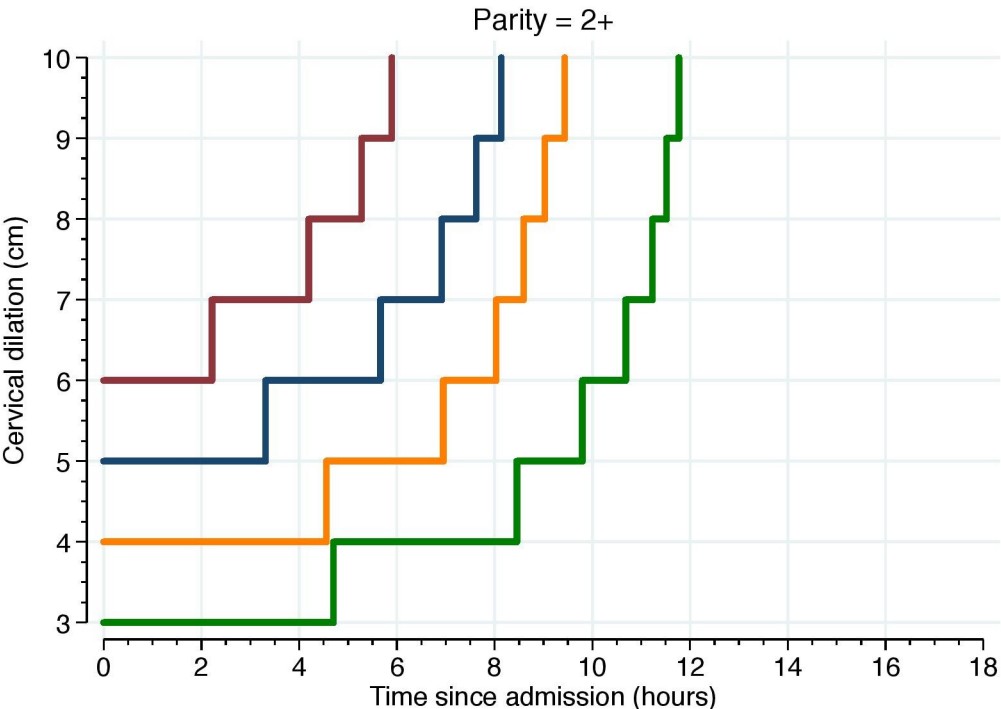

**Fig 7. The 95<sup>th</sup> percentiles of cumulative duration among parity = 2.** Illustrate the 95th percentiles of cumulative labor duration in parity = 2+ and plotted as staircase lines based on the cervical dilation observed at admission, and centimeter by centimeter until full cervical dilation. The staircase cumulative duration is equal to the 95th percentiles in Table 3.

## Clinical management and setting

The variety of duration of first stage shown in this study and compared with the results by Zhang [26] and Oladapo [23], could reflect differences in obstetrical settings, clinical management factors and possibly, to a minor extent, differences in maternal and neonatal anthropometrics, such as or example BMI and fetal weight. Further, due to the strict inclusion criteria in this study, with 100% effacement, it could be that more women are in advanced first stage of labor when admitted compared to the women in Zhangs cohort [26]. More than 80% of the women in the Oladapo study had advanced cervical effacement (very thin cervix) [23]. The

**Table 4. Duration of labor compared with Zhang [26] and Oladapo[23] results.**

| Cervical dilation | Cervical dilation | Current study | Zhang et al | Oladapo et al |
|---|---|---|---|---|
| 3 | 4 | 0.83 (0.14–4.97) | 1.8 (8.1) | 2.82 (0.60–13.33) |
| 4 | 5 | 0.89 (0.15–5.26) | 1.3 (6.4) | 1.72 (0.38–7.83) |
| 5 | 6 | 0.68 (0.10–4.55) | 0.8 (3.2) | 1.19 (0.23–6.17) |
| 6 | 7 | 0.48 (0.06–3.82) | 0.6 (2.2) | 0.66 (0.09–4.92) |
| 7 | 8 | 0.33 (0.03–3.62) | 0.5 (1.6) | 0.25 (0.02–3.10) |
| 8 | 9 | 0.24 (0.02–3.00) | 0.5 (1.4) | |
| 8 | 10 | | | 0.87 (0.18–4.19) |
| 9 | 10 | 0.18 (0.01–2.66) | 0.5 (1.8) | |

Data are reported as median hours (5<sup>th</sup>, 95<sup>th</sup> percentiles)

Zhang et al data reported as median hours (95<sup>th</sup> percentile)

few known causal factors influencing first stage duration are maternal age, increasing BMI, posterior fetal position and higher fetal weight [34–40]. Compared to the African and American cohorts, the Swedish cohort was older, had lower BMI, lower gestational age at birth, and fetal weight was higher. Strengths and regularity of contractions have not been evaluated in any of these studies but could influence the patterns of labor.

The influence of epidural analgesia on the duration of first stage of labor remains unclear and was not evaluated in this study [41]. Our cohort had a higher rate of oxytocin (49% in nulliparous) than the two other cohorts (47% Zhang, 40% Oladapo). Oladapo's results demonstrated slightly faster labor progression when excluding women with oxytocin, the authors emphasized the translational relevance of including women with augmentation due to the frequent use [23]. Because synthetic oxytocin use is common in Swedish maternity care, we choose to include those receiving this intervention to avoid selection bias and mirror the clinical setting.

## Why is labor progression and duration important in childbirth?

One reason to reevaluate normal labor progression is to facilitate understanding of normal birth. This is critical to building maternity care systems that endorse patience when labor progress is normal and intervention only when it is maximally beneficial. This could lead to reduction of unnecessary caesarean sections. The LAPS study compared the WHO partograph with the Zhang guidelines on normal progression [42]. The hypothesis was that a more dynamic labor curve, allowing more time early in labor, before 5–6 cm, would affect the caesarean delivery rate. Result of this study showed an overall decrease in the cesarean delivery rate for all women within the LAPS study, regardless of which guideline was used. The authors suggest that the overall lower cesarean rate during the study period was related to an increased focus on labor progression, rather than the use of a new guideline [42]. The definition of prolonged labor in the intervention group was based on the traverse time in Zhangs cohort from women with the slowest labors (95th percentile) (2.2, 1.6, 1.4, 1.8) [26]. The corresponding traverse time 95th percentile (3.82, 3.62, 3.00, 2.66) in our study suggests even wider time intervals throughout the first stage of labor including progression at and beyond 6 centimeters of cervical dilation. Therefore, we speculate that Zhang's definition of prolonged labor after 5–6 centimeters dilation might be too conservative in some settings.

## Labor progression for all women (including cesarean delivery)

One major challenge in studies on labor progression is how to avoid potential selection bias in the cohort. To exclude women giving birth by cesarean and/or with adverse neonatal outcome might introduce selection bias. To investigate this, we created a target population cohort including women regardless of mode of delivery and regardless of neonatal outcome (S1–S3 Tables, S1–S4 Figs). In the supportive material we have presented a target population cohort of 101,730 women and our findings suggest that there are important differences between the cohorts that needs to be further explored. Interestingly, fetal posterior position was almost twice as common when women with cesarean delivery were included in the cohort. For instance, nulliparous women in this cohort had two hours longer duration at the 95th percentile when we included those who delivered by CS vs. findings when those with CS and adverse neonatal outcome were excluded. These results correspond with findings of the LAPS Study which showed that women who delivered by intrapartum cesarean delivery had a prolonged labor duration compared with those who delivered vaginally in both study groups [43]. Future research is needed to better understand if labor duration or other factors, such as fetal malposition, may signal risk for cesarean delivery and/or adverse neonatal outcome.

## Clinical implications

Our findings, considered in conjunction with Zhang's [26] and Oladapo's [23] work, indicate that greater patience with first stage labor progress is warranted and suggest the need for increased clinical and hospital policy focus on all aspects of progression in labor. Average first stage of labor progression expectations based on the traditional partograph should not be used for clinical management. Based on these cumulative findings, a line of inquiry aimed to revise labor dystocia definitions is necessary.

## Strengths and limitations

There are several study strengths. First, we had access to a large cohort of more than 85,000 women contributing information on progress of labor. As women are admitted to the hospital at different stages of labor (from 3 to 6 cm cervical dilation) the large size of the cohort permits robust analysis for estimating labor durations in a wide variety of clinical scenarios, including those who were admitted to the hospital at earlier as well as more advanced labor. Furthermore, by using the unique personal identity number all information was collected prospectively in standardized antenatal obstetric and neonatal records, minimizing the potential for recall bias [44]. Selection bias is further minimized as all pregnant women in Sweden are offered free health care from pregnancy to postpartum care and more than 99% of all women attend antenatal care and give birth in a hospital. The cohort consists of approximately 25% of all births in Sweden and the unique equal health care system available for all women strengthens the external validity as this is a cohort based on the average Swedish population. While we acknowledge the limitation of describing 'normal' labor patterns in a sample with frequent intervention, we also believe that the inclusion of women with interventions such as amniotomy, epidural, and oxytocin augmentation strengthens the generalizability of our results to current obstetric practice. Robust consideration of the appropriate methodological approach to advancing this science that have been published during the last decade and directly informed our analysis [29, 45]. By using strict criteria (C1-C2) for inclusion in the cohort we minimized potential selection bias arising due to inclusion of women admitted to the hospital before the first stage of labor. Because the inclusion criteria included several clinically validated parameters we consider it to be a model with high precision. It has been almost a decade since Zhang presented the labor curves on an American population and a number of smaller studies have been published using the same statistical approach. This is the first cohort study based on a Swedish population and, to our knowledge, it is the labor progression study with the largest sample. The landmark research by Friedman enormously impacted labor management over the last 60 years. To refine and update our understanding of normal labor progress and labor management, more research using contemporary cohorts is necessary. Our study, using the same statistical approach as seminal researchers and analyzing a new cohort, contributes to this broader effort.

There are also limitations to this study. We excluded women with induced labors, cesarean deliveries, and pregnancies in non-vertex presentation to enable comparison to seminal findings, therefore our findings cannot be generalized to these women. Also, women are admitted to the hospital at different stages of labor, and they may differ in terms of labor progression and clinical management. It could be speculated that women who are admitted early during first stage may be exposed to more interventions (amniotomy or oxytocin) to speed up progression compared to women who are admitted later during first stage, which has not been evaluated in this study. The start of contractions among spontaneous onset labors is at home and the starting point of regular painful contractions could be prone to recall bias. Since this time point is one of several criterions we still believe it is a model of high precision. In future

studies on "normal progress of labor" maternal outcomes such as postpartum hemorrhage, perineal lacerations, urinary and faecal incontinence, and negative birth experience may also be considered being included in the definition "adverse birth outcome". Importantly, normal delivery outcomes were restricted to infant outcomes and we cannot conclude anything about the risk for adverse maternal outcomes in relation to labor duration. Finally, the supportive material in this study revealed important differences in duration between the target population cohort and the study cohort. This identifies the need to evaluate differences in progression of labor and labor duration in various cohorts to identify the threshold for true dystocia.

## Conclusion

Our findings, considered in conjunction with Zhang's [26] and Oladapo's [23] work, indicate that greater patience with first stage labor progress is warranted and suggest the need for increased clinical and hospital policy focus on all aspects of progression in labor. Average first stage of labor progression expectations based on the traditional partograph should not be used for clinical management. This large cohort study of first stage of labor indicates that duration and progression differ substantially from the traditional partograph alert lines in defining normal progress of labor. Similar to other recent labor progress research, our results show an acceleration of cervical advancement beginning at 5–6 centimeters of dilation. Swedish women's cervical dilation can progress both faster and significantly slower than 1 cm per hour throughout the first stage of labor and still conclude in vaginal birth with normal neonatal outcomes. These results, considered with findings of other first stage of labor progress research, signal the need for a line of inquiry aimed to revise labor dystocia definitions.

To successfully distinguish thresholds for when labor is too long for the individual woman and infant, we need to gain more knowledge on both protective factors and risk factors. Future research might be enhanced through evaluating long-term follow up on infant and maternal health in relation to labor progress and duration estimates of both the first and second stages of labor duration. Given our findings that traverse time is slower when women with cesarean delivery and adverse neonatal outcome are included in the analysis, we also recommend that future analyses consider both observations with vaginal birth and normal outcomes as well as cesarean birth and abnormal outcomes. Duration of time in labor has long been considered of central importance in distinguishing risk. Our findings in the context of similar results, demonstrate the wide variability of total first stage of labor durations and trajectories of cervical dilation among those birthing vaginally with normal neonatal outcomes. This may signal that factors other than duration of the first and second stages of labor are more critical to maternal/child risk.

## Supporting information

**S1 Table. Maternal, labor and fetal characteristics for target population cohort.**
(DOCX)

**S2 Table. Duration of labor in hours from one-centimeter dilation to the next, by parity for target population cohort.**
(DOCX)

**S3 Table. Cumulative duration of labor in hours in para 0, 1, and 2+ based on the cervical dilation at admission for target population cohort.**
(DOCX)

**S1 Fig. Diagram of patient selection for target population cohort.**
(TIF)

**S2 Fig. The 95th percentiles of cumulative duration among nulliparous for target population cohort.**
(TIF)

**S3 Fig. The 95th percentiles of cumulative duration among parity = 1 for target population cohort.**
(TIF)

**S4 Fig. The 95th percentiles of cumulative duration among parity = 2+ for target population cohort.**
(TIF)

## Author Contributions

**Conceptualization:** Louise Lundborg, Anna Sandström, Olof Stephansson, Mia Ahlberg.

**Methodology:** Louise Lundborg, Andrea Discacciati, Ellen L. Tilden, Olof Stephansson.

**Project administration:** Louise Lundborg, Olof Stephansson, Mia Ahlberg.

**Resources:** Mia Ahlberg.

**Software:** Louise Lundborg.

**Supervision:** Ellen L. Tilden, Olof Stephansson, Mia Ahlberg.

**Validation:** Louise Lundborg, Andrea Discacciati, Olof Stephansson, Mia Ahlberg.

**Writing – original draft:** Louise Lundborg, Andrea Discacciati, Mia Ahlberg.

**Writing – review & editing:** Louise Lundborg, Katarina Åberg, Anna Sandström, Ellen L. Tilden, Olof Stephansson.

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
