## [Decision Letter · Decision Letter 0]

29 May 2020

PONE-D-20-06618

First stage progression in women with spontaneous onset of labor: A large population-based cohort study

PLOS ONE

Dear Dr. Lundborg,

Please see my additional comments, below. - For minor revision. RCY

Thank you for submitting your manuscript to PLOS ONE. After careful consideration, we feel that it has merit but does not fully meet PLOS ONE’s publication criteria as it currently stands. Therefore, we invite you to submit a revised version of the manuscript that addresses the points raised during the review process.

We look forward to receiving your revised manuscript.

Kind regards,

Roger C. Young

Academic Editor

PLOS ONE

Journal Requirements:

2. In ethics statement in the manuscript and in the online submission form, please provide additional information about the database used in your retrospective study. Specifically, please ensure that you have discussed whether all data were fully anonymized before you accessed them and/or whether the IRB or ethics committee waived the requirement for informed consent. If patients provided informed written consent to have their data used in research, please include this information.

Additional Editor Comments (if provided):

The reviews place this in the "minor revision" category, but actually some of the questions raised may require a bit of re-writing.

Please respond to all reviewers' questions comments and questions, but both reviewers mention the censoring effects of C/S - please give this question extra attention.

Reviewers' comments:

Reviewer's Responses to Questions

**Comments to the Author**

1. Is the manuscript technically sound, and do the data support the conclusions?

Reviewer #1: Yes

Reviewer #2: Yes

2. Has the statistical analysis been performed appropriately and rigorously? 

Reviewer #1: Yes

Reviewer #2: Yes

3. Have the authors made all data underlying the findings in their manuscript fully available?

Reviewer #1: Yes

Reviewer #2: Yes

4. Is the manuscript presented in an intelligible fashion and written in standard English?

Reviewer #1: Yes

Reviewer #2: Yes

5. Review Comments to the Author

Reviewer #1: Thank you for the opportunity to review this excellent and significant manuscript. It adds another very large analysis to the obstetrical literature from another geographical region, describing the relationship between dilation and time using 3 different techniques. It confirms what all other recent studies have shown namely that, on average, dilation proceeds in a near exponential fashion not the sigmoid shape depicted in the traditional Freidman curves. In addition, the authors confirm the very wide variation in dilation rates and clearly state the problem that this large variation creates when one tries to use an “average” curve to manage labor. This conclusion is very important because generations of Obstetricians have been taught to evaluate labor based on a presumed average curve with relatively narrow variation in the rates of expected progress. Although there is now wide acceptance that the dilation curve is not sigmoid in shape, nor linear after 3 cm, there is less appreciation that there is no substantial clustering (central tendency) of patients around an “average”. That is, while it is always possible to contruct a mathematical average, few patients actually follow this mathematical average. Bravo for stating this so explicitly and repeatedly throughout this paper.

In addition, results indicated that NTSV Swedish women progressed faster in early first stage compared to findings the in USA based publications (Zhang 2010) and faster compared to 2 subsaharan countries women (Oladapo 2018). In late first stage, Swedish women progressed more slowly than the rates reported for USA women and similar to those reported for the Ugandan and Nigerian women.

There are a few points worth mentioning in the discussion.

Developing a model with precision:

The general problem of modeling a process (in this case, cervical dilation) with precision: The authors have demonstrated excessively wide variation when the model is based on only parity and time. It would be useful to discuss the known causes of poor modeling precision such as imprecise measurements and unmeasured influential factors. The observation that model precision improved when the models are restricted to higher and higher admission dilations suggest that some unmeasured factors may become more homogeneous with advancing labor. Cervical effacement, contraction strength and frequency, epidural use and censoring due to intrapartum cesarean are candidates.

Entry point and the effect of incomplete effacement:

Re comparisons of traverse time at low dilations. You noted a faster median time to advance from 3 to 4 cm in the Swedish women 0.83h (95th 4.97) compared to USA 1.8h( 95th 8.1) and compared to 2 subSaharn African countries 2.8h (95th 13.3) I presume these are Uganda and Nigerian women based on other more detailed publications by Oladapo et al. The inclusion criteria in this Swedish study were women with singleton, term, vertex presentations and spontaneous onset of labor and vaginal birth. Exclusion for < 3 examinations, missing exam at 10 cm, Apgar 5 min <7, or any of diagnosis related to HIE or meconium aspiration.

The onset of active labor was defined as the presence of regular painful contractions and a well effaced cervix and a dilation of 3cm or rupture of membranes with a dilation of 3 cm.

Zhang et al 2010 included NTSV patients with spontaneous onset of labor without a requirement for a specific dilation (10% had a dilation of 1 cm or less) or a requirement for a specific effacement ( 10% had an admission effacement of 60% or less.

Please present the comparative statistics on the admission examination findings because the state of the cervix on admission has a large influence on rates of dilation in the beginning.

Censoring effect of cesarean:

The NTSV cesarean rates in Sweden are much lower than in the USA and perhaps more comparable to that in the Oladapo studies.

In the LAPS study, Swedish CS rates in NTSV in 2014 were approximately 9.5% before the intervention.

In the USA CS rates for NTSV were approximately 27.5% in 2007 (Zhang et al 2010, women mostly from 2005- 7)

Please comment on how the removal of more patients for labor disorders could affect the dilation rates in the residual population used for modeling. I agree that maternal and fetal anthropomorphic differences can account for some differences but this censoring effect could be present as well and bears mentioning.

Recommendations for clinical practice drawn from a study of “normal” labors.

You have commented very carefully on how labor curves and your findings could help direct management. I think it would be very useful to remind the reader that any study that intentionally excludes all abnormal outcomes cannot determine the safety of a particular intervention threshold.

Reviewer #2: This paper used a large electronic obstetric database to examine the labor pattern and duration of first stage of labor. The analysis was well done and the paper clearly written. The findings are reasonable and potentially useful clinically. To improve this paper, several issues need more attention by the authors:

1. What was the overall CS rate in this study population? What was the proportion of prelabor vs. intrapartum CS? Given that the overall CS rate in Sweden (18%?) is low and intrapartum CS rate is even lower, the observed longer duration of labor at the 95th percentile than that in the Zhang's data appears reasonable as the U.S. overall CS rate is over 30%. This could be one of the explanations for the difference between the studies.

2. Ideally, the following women should also be excluded from the study: women with a scarred uterus, postpartum hemorrhage or III or IV degree perineal laceration. This may also address the issue stated on Line 180-181 in Strength and limitation.

3. The authors defined "the first stage of labor" and "onset of active phase" but in the paper, they never really showed any results that were associated with these two definitions. If it is true, then I'd suggest not to mention them as these definitions are still controvercial.

4. The median cervical dilation at admission was 5 cm. Thus, the admission seems a little late, comparing to other databases. What was the median effacement?

5. The number of vaginal exams tended to be at low side. The authors may want to discuss the impact of fewer data points on the results.

6. Women who were admitted at different phases of labor may differ in terms of labor progression. How that may have affected the results? Some discussion is needed.

6. PLOS authors have the option to publish the peer review history of their article (what does this mean?). If published, this will include your full peer review and any attached files.

Reviewer #1: No

Reviewer #2: No

---

## [Author Response · Author response to Decision Letter 0]

3 Aug 2020

Dear Editor

Thank you for your thoughtful critique of our manuscript ‘First stage progression in women with spontaneous onset of labor: A large population-based cohort study’. We have made several changes to the manuscript in response to your critique, and we feel that these changes strengthen the manuscript.

Of note, we have given the censoring effect of CS extra attention as you and the two reviewers requested. The scope of this study included normal labors for comparison to seminal findings. However the question of selection bias by excluding cesarean delivery (and adverse neonatal outcome) is a problem; we agree with the reviewers comments and have adressed them. To address the question of selection bias and censoring effect of excluding CS and births with adverse neonatal outcomes we have now added additonal analyses on labor duration in a target population cohort including women delivered by CS and adverse neonatal outcomes. Results of these additional analyses have been added to supplemental materials.

Please see the questions and answers (in italics) together with line numberings (in red) corresponding to the changes made in the tracked changes document. 

Reviewer 1#

1. Developing a model with precision

The general problem of modeling a process (in this case, cervical dilation) with precision: The authors have demonstrated excessively wide variation when the model is based on only parity and time. It would be useful to discuss the known causes of poor modeling precision such as imprecise measurements and unmeasured influential factors. The observation that model precision improved when the models are restricted to higher and higher admission dilations suggest that some unmeasured factors may become more homogeneous with advancing labor. Cervical effacement, contraction strength and frequency, epidural use and censoring due to intrapartum cesarean are candidates.

We consider the large cohort that we used for this analysis to be a major strength related to this concern. The size of the sample provides enough obeservations to enable robust point estimates even at smaller dilation. We also consider the fact of the strict inclusion criteria- e.g., only including women with painful regular contactions, fully effaced cervix, and a notation of cervical dilation- as a strength on this concern as these increase the likelihood that we only included women who are in active labor and not in the latent phase. This has been clarified in the Strengths and Limitations section (lines 461-468) (lines 479-486).

This approach differs from the studies by Zhang and Oladapo and these differences are now clarified in the Discussion (lines 364-367).

2. Entry point and the effect of incomplete effacement

Re comparisons of traverse time at low dilations. You noted a faster median time to advance from 3 to 4 cm in the Swedish women 0.83h (95th 4.97) compared to USA 1.8h( 95th 8.1) and compared to 2 Sub Saharian African countries 2.8h (95th 13.3) I presume these are Uganda and Nigerian women based on other more detailed publications by Oladapo et al. The inclusion criteria in this Swedish study were women with singleton, term, vertex presentations and spontaneous onset of labor and vaginal birth. Exclusion for < 3 examinations, missing exam at 10 cm, Apgar 5 min <7, or any of diagnosis related to HIE or meconium aspiration.

The onset of active labor was defined as the presence of regular painful contractions and a well effaced cervix and a dilation of 3cm or rupture of membranes with a dilation of 3 cm.

Zhang et al 2010 included NTSV patients with spontaneous onset of labor without a requirement for a specific dilation (10% had a dilation of 1 cm or less) or a requirement for a specific effacement ( 10% had an admission effacement of 60% or less.

Please present the comparative statistics on the admission examination findings because the state of the cervix on admission has a large influence on rates of dilation in the beginning.

Cervical effacement, one of the criterion in this study, was 100% (or almost) effacement. We have now clarified this in the defintion on starting point for first stage in the method section (lines 155-170). We have also added information regarding self-report issues (contractions) and clinical examination in both the Methods section and in Strengths and Limitations (lines 155-170 ) (lines 479-486).

3. Censoring

Please comment on how the removal of more patients for labor disorders could affect the dilation rates in the residual population used for modeling. I agree that maternal and fetal anthropomorphic differences can account for some differences but this censoring effect could be present as well and bears mentioning.

Thank you for this important critque. To explore the censoring effect of cesarean delivery we conducted the same analyses in the whole population (target population), including all intrapartum CS as well as adverese neonatal outcomes. (lines 172-180) We have added results in supportive material. We have also expanded the discussion to comment these findings. (lines 309-326) (lines 410-426) (lines 511-512)

4. Safety

You have commented very carefully on how labor curves and your findings could help direct management. I think it would be very useful to remind the reader that any study that intentionally excludes all abnormal outcomes cannot determine the safety of a particular intervention threshold.

We agree. We have now updated and clarified the limitations of excluding abnormal outcomes and reminded the reader about this limitation in the Strengths and Limitations sections. (lines 486-489) (lines 511-514)

Reviewer # 2:

1. What was the overall CS rate in this study population? 

Cesarean delivery 11.4 % for nulliparous, 2.1% for parity 1 and 1, 6% for parity 2+. ( S.Table 1)

2.What was the proportion of prelabor vs. intrapartum CS? 

Those with prelabor cesarean deliveries were not included in this study. The intrapartum cesarean delivery rates within this cohort are 11.4 % for nulliparous, 2.1 % for parity 1, and 1.6% for parity 2+.( S.Table 1)

3. Given that the overall CS rate in Sweden (18%?) is low and intrapartum CS rate is even lower, the observed longer duration of labor at the 95th percentile than that in the Zhang's data appears reasonable as the U.S. overall CS rate is over 30%. This could be one of the explanations for the difference between the studies.

We agree that this variation in CS rates is important and should be further addressed indicate. To our discussion under the headline “Why is labor progression..”. (lines 389-409), we have now added statistical analysis for a target population cohort to further fully describe the effects of labor duration of including/excluding women due to cesarean delivery and adverse neonatal outcomes. (lines 172-182) (lines 313-330) (lines 410-426) (lines 511-514)

4. Ideally, the following women should also be excluded from the study: women with a scarred uterus, postpartum hemorrhage or III or IV degree perineal laceration. This may also address the issue stated on Line 180-181 in Strength and limitation.

Women with scarred uterus (previous cesarean) are already excluded in original manuscript in this study cohort. (Fig 1, line 133). We agree that women with postpartum hemorrhage or III or IV degree perineal laceration should be excluded when evaluating normal labors, however for comparison to seminal findings we decided to keep the cohort as identical as possible to the Zhang and Oladapo cohorts. For future research on labor progression, ideally women with postpartum hemorrhage or 3,4-degree perineal laceration should be excluded - we agree with you on this point. This is now stated in Strengths and limitations. (lines 486-492)

5. The authors defined "the first stage of labor" and "onset of active phase" but in the paper, they never really showed any results that were associated with these two definitions. If it is true, then I'd suggest not to mention them as these definitions are still controvercial.

We have now changed the phrasing “onset of active phase” to ‘starting point for first stage of labor’, which we agree is a better definition. Furthermore, we have now clarified the definition of starting point for duration of first stage of labor in this study in the Methods section. (lines 155-170). 

6. The median cervical dilation at admission was 5 cm. Thus, the admission seems a little late, comparing to other databases. What was the median effacement?

From our review of the literature, we note that the median cervical dilation at admission differs in different cohorts. This likely reflects how care is organized in different countries or potentially differences in women’s expectations of when to seek hospital admission for birth. Our study is based on prospectively collected data by certified midwives, and we consider the data to be of high precision. 

 The criteria for inclusion in this study was complete or nearly 100 % effacement meaning that there was no median to report. (lines 155-170)

7. The number of vaginal exams tended to be at low side. The authors may want to discuss the impact of fewer data points on the results.

We have discussed the question of precision (fewer data points) in the original manuscript under strenghts and limitations. (lines 461-468) (lines 479-486). (lines 443-448) (Please see answer number 1 rewiver 1). Due to the size of the cohort as well as the large portion of the sample with estimates for traverse times, we believe that our estimates on this point are robust. (Table 2, Please see numbers of women included for every cm dilation). The number of vaginal exams [median 4 (2,7)] reflects how midwives and clinicans work in the study setting. These clinical practices are similar to the number of vaginal exams found in prior research: 3 (2,5) (Oladapo), 5 (1, 9) (Zhang). 

8. Women who were admitted at different phases of labor may differ in terms of labor progression. How that may have affected the results? Some discussion is needed.

We aimed to avoid including women admitted too early in labor ( possibly in the “latent phase”), and we have added additional information to address the above question under strengths and limitations. (lines 461-464) Also, as the reviewer helpfully noted in question 5, the definiton of “active phase” is still controversial. We have therefore removed this expression from the manuscript. ( Please see answer number 5 rewiver 2)

However, it could be speculated that women admitted early vs. late during first stage differ in terms of labor progression as shown in cumulative distributions. Thus we agree with you on this point. The impact of more/ less interventions in early vs late admission was not evaluated in this study. This is now more clearly stated in strengths and limitations. (lines 479-486)

---

## [Decision Letter · Decision Letter 1]

4 Sep 2020

PONE-D-20-06618R1

First stage progression in women with spontaneous onset of labor: A large population-based cohort study

PLOS ONE

Dear Dr. Lundborg,

Thank you for submitting your manuscript to PLOS ONE. After careful consideration, we feel that it has merit but does not fully meet PLOS ONE’s publication criteria as it currently stands. Therefore, we invite you to submit a revised version of the manuscript that addresses the points raised during the review process.

Please see my comments below.

We look forward to receiving your revised manuscript.

Kind regards,

Roger C. Young

Academic Editor

PLOS ONE

Additional Editor Comments (if provided):

This revision is significantly improved. I would like you to consider the following suggestions. To emphasize, these are suggestions only and I leave open the opportunity for you to not include them in the final version if you so choose.

Abstract:

Results:

1. ,[the rate of] cervical dilation accelerates ...

2. Among nulliparous women, the median time [found in our population] was faster [than] their counterparts...

3. Among nulliparous and multiparous women [our data] suggest that ... [, however, ] traverse time at and beyond.....is [slower] than 1 cm per hour.

Conclusions:

1. ...widely and [labors experiencing a prolonged first stage] can still...

2. 1 cm [per hour cervical dilation rate for the first stage of labor] may not be [universally] meaninful.

3. ... differences in progression[for women] in different populations.

4. [For] prolonged labor progression [to be more clinically meaningful, the] association with adverse birth outcomes needs to be further investigated [in specific populations].

Reviewers' comments:

Reviewer's Responses to Questions

**Comments to the Author**

1. If the authors have adequately addressed your comments raised in a previous round of review and you feel that this manuscript is now acceptable for publication, you may indicate that here to bypass the “Comments to the Author” section, enter your conflict of interest statement in the “Confidential to Editor” section, and submit your "Accept" recommendation.

Reviewer #2: (No Response)

2. Is the manuscript technically sound, and do the data support the conclusions?

Reviewer #2: Yes

3. Has the statistical analysis been performed appropriately and rigorously? 

Reviewer #2: Yes

4. Have the authors made all data underlying the findings in their manuscript fully available?

Reviewer #2: Yes

5. Is the manuscript presented in an intelligible fashion and written in standard English?

Reviewer #2: Yes

6. Review Comments to the Author

Reviewer #2: (No Response)

7. PLOS authors have the option to publish the peer review history of their article (what does this mean?). If published, this will include your full peer review and any attached files.

Reviewer #2: No

---

## [Author Response · Author response to Decision Letter 1]

10 Sep 2020

Thank you for your response to our resubmission on the manuscript ‘First stage progression in women with spontaneous onset of labor: A large population-based cohort study’. 

We have made final minor changes in the abstract, only according to your thoughtful last suggestions. No other changes have been made in the manuscript. 

Thank you for your attention to our paper.

---

## [Editor Report · Decision Letter 2]

14 Sep 2020

First stage progression in women with spontaneous onset of labor: A large population-based cohort study

PONE-D-20-06618R2

Dear Dr. Lundborg,

We’re pleased to inform you that your manuscript has been judged scientifically suitable for publication and will be formally accepted for publication once it meets all outstanding technical requirements.

Kind regards,

Roger C. Young

Academic Editor

PLOS ONE

---

## [Editor Report · Acceptance letter]

16 Sep 2020

PONE-D-20-06618R2 

First stage progression in women with spontaneous onset of labor: A large population-based cohort study 

Dear Dr. Lundborg:

I'm pleased to inform you that your manuscript has been deemed suitable for publication in PLOS ONE. Congratulations! Your manuscript is now with our production department. 

Kind regards, 

on behalf of

Dr. Roger C. Young 

Academic Editor

PLOS ONE